# Assessment of the Influence of Storage Conditions and Time on Red Currants (*Ribes rubrum* L.) Using Image Processing and Traditional Machine Learning

Ewa Ropelewska 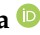

Fruit and Vegetable Storage and Processing Department, The National Institute of Horticultural Research, Konstytucji 3 Maja 1/3, 96-100 Skierniewice, Poland; ewa.ropelewska@inhort.pl

**Abstract:** This study was aimed at revealing the usefulness of the combination of image analysis and artificial intelligence in assessing the quality of red currants in terms of external structure changes under the influence of different storage conditions. Red currants after harvest were subjected to storage at room temperature and at a lower temperature in the refrigerator for one week and two weeks. The statistically significant differences in selected image textures as a result of prolonged storage were determined for both samples stored in the room and the refrigerator. However, the changes in the structure of the red currant samples stored at room temperature were greater than for storage in the refrigerator. Distinguishing samples using models built using machine learning algorithms confirmed the usefulness of selected textures to assess the influence of storage conditions and time on red currants. Unstored red currants, samples stored at room temperature for one week, and those stored at room temperature for two weeks were classified with an accuracy of 99–100%, and unstored samples, fruit stored in the refrigerator for one week, and that stored in the refrigerator for two weeks were correctly distinguished at an accuracy of 97–100%, depending on the algorithm. Models developed for distinguishing red currants stored at room temperature and in the refrigerator for one week provided an accuracy of 99–100%, and for the classification of red currants stored at room temperature and in the refrigerator for two weeks, an accuracy equal to 100% for all used algorithms was determined.

**Keywords:** stored red currants; room temperature; refrigerator; digital imaging; artificial intelligence

## 1. Introduction

The genus *Ribes* belonging to the Grossulariaceae family includes mainly deciduous or semievergreen shrubs. The species of *Ribes* are widely distributed in temperate and cold regions of the Northern Hemisphere, such as northern Europe, and northern and central Asia [1]. From the genus *Ribes* including more than 150 species, only a few species are cultivated for red currant production. Red currant (*Ribes rubrum* L.) is native to Western Europe. It is cultivated in both commercial plantations and home gardens in regions with moderate temperatures. Poland is one of the main producers of currants. The cultivation of currant is not demanding and gives producers high profitability [2,3]. Currant is a commercially important but relatively young crop. The first red currant crops appeared at the beginning of the 15th century, and it was widely cultivated in gardens in the 16th century [4]. Red currant has become popular and spread as a result of the demand for berries due to their high therapeutic effect, as well as technological efficiency, and economically valuable characteristics [5]. Red currant provides healthful and delicious fruits which are a rich source of organic acids, vitamins, and phenolics with antioxidant and antiradical properties [6,7]. Red currant is characterized by the presence of several important phenolic compounds, such as gallic acid, rutin, syringic acid, cinnamic acid, (+)-catechin, ferulic acid, and chlorogenic acid [8]. Due to the presence of phenolics, intakes of red currant may be associated with a reduced risk of, among others, heart disease, cancer, and stroke [6].

Currants can be an ingredient in functional foods or dietary supplements [9]. Red currant can be consumed fresh and in processed forms in jam, marmalade, jelly, ice cream, fruit juice, or dried fruit [6].

Currants are soft berries. Although organoleptic characteristics of fresh berries are generally accepted by consumers and can be successful in the fresh market, their shelf life is usually short [10]. Red currants are also susceptible to microbial spoilage [11]. Defects in fruit during the storage process can also be the result of physiological disorders or mechanical damage [12]. Fresh fruits are highly perishable, and different pre- and postharvest factors can affect their quality and shelf life during storage [13]. Often, raw materials and their products are stored in a frozen state or at a low temperature. Due to the need to extend the shelf life of currants through proper storage or processing, there is also a need to assess the quality of fruit subjected to these processes [14]. Food product quality mainly based on color and firmness can be important both to consumers and trade. Currants after harvest may be exposed to changes in firmness caused by moisture loss over time depending on storage temperature. Firmness can be related to structuring material, cell turgor, or shape and size of cells. Color changes can be caused by the tissue structure [15]. Relationships between firmness, tissue microstructure, and optical properties during storage were confirmed and it was reported that optical properties can be used to evaluate the changes in fruit microstructure during post-harvest storage [16,17].

Machine vision involving imaging techniques can be useful in detecting external quality parameters of fruit [18]. Generally, the importance of machine vision in agriculture has increased, including non-destructive quality control, inspection of external features of fruit, and classification based on texture, color, shape, size, and presence of damage. The changes in textures can be related to color differences and can be used to detect external defects [19]. Machine vision involving color image processing can provide high classification accuracy and allow the detection of slight changes in an objective, inexpensive, easy, and fast manner. Image texture as a function of spatial variation in pixel values can provide numerical data from the image of the object and can even specify the changes difficult to perceive visually [20–22]. The combination of imaging and machine learning can be used to monitor the changes in fruit quality throughout storage [23]. Artificial intelligence can support decision-making to predict the highest quality of fruit and define sales strategies [24]. The previous literature data indicated high effectiveness of various deep learning and traditional machine learning algorithms to classify samples with an accuracy reaching 100% [25–28].

The objective of this study was to reveal the usefulness of the combination of image analysis and artificial intelligence in assessing the quality of red currant in terms of external structure changes under the influence of different storage conditions. The applied procedure is a great novelty for the detection of fruit quality during storage. Due to the use of image features from different color channels *R*, *G*, *B*, *L*, *a*, *b*, *X*, *Y*, and *Z* to build models using machine learning algorithms, fruit quality monitoring was carried out in a non-destructive, objective, and effective manner.

## 2. Materials and Methods

### 2.1. Material

The red currants were collected from the local garden located in northeastern Poland. Red currants have been grown in the garden for several years. This study generally aimed to demonstrate the usefulness of image analysis and artificial intelligence in assessing the quality of stored red currant. Therefore, one random cultivar was used in the experiments without considering the characteristics and genetic origin of this cultivar. The only visual criterion for selecting a cultivar was large, fully developed fruit in a given growing season. The red currants were harvested at the stage in which fruit were richly colored, juicy, and firm. These features were assessed organoleptically. From each bunch of red currants, several undamaged fruits with stems were sampled, to obtain a total of 400 fruits. In the storage experiments, fruit with stems was used so that the structure of the fruit was not

damaged during the removal of fruit from the bunches. Individual fruits separated from the bunches were used to ensure the same storage conditions for each fruit. Therefore, red currants were stored as a single layer of fruit. Fruit storage was carried out in plastic boxes with perforated walls. The obtained sample was divided into two parts of 200 red currants intended for storage in different conditions. The first part of the 200 fruits was stored at a room temperature of $23 \pm 1$ °C and the second part of the 200 fruits was placed in the refrigerator (Beko, Istanbul, Turkey) at a temperature of 4 °C. Unstored samples, both intended to be stored at room temperature and in a refrigerator, were imaged. Then, the same samples were imaged every week. After two weeks, when very distinct changes in the appearance (size and shape changes such as wrinkling, visible losses of mass and water, and color changes such as fruit darkening) of fruit stored in the room were noticed, indicating complete damage to the structure of most of the red currants, the experiment was stopped.

### 2.2. Image Analysis

The unstored red currants and then stored for one week and two weeks were imaged using a digital camera (Auto White Balance, Optical Image Stabilization, 8x digital zoom, F 2.4) and lighting (24 LED, Related Output Power of 2.2 W, Related Input current of 0.07 A, Related Input Voltage of AC110-240 V/50–60 Hz). Color calibration of the digital camera was carried out. Imaging was performed in a box. Red currants were placed individually on a background so as not to touch each other. Fifty objects were included in each of the images. Red currant images were obtained under room conditions. In total, the following were acquired:

- 200 imaged unstored red currants intended for storage at room temperature;
- 200 imaged red currants stored at room temperature for one week;
- 200 imaged red currants stored at room temperature for two weeks;
- 200 imaged unstored red currants intended for storage in the refrigerator;
- 200 imaged red currants stored in the refrigerator for one week;
- 200 imaged red currants stored in the refrigerator for two weeks.

The images of red currants directly after harvest are presented in Figure 1. Whole undamaged fully colored red currants are visible.

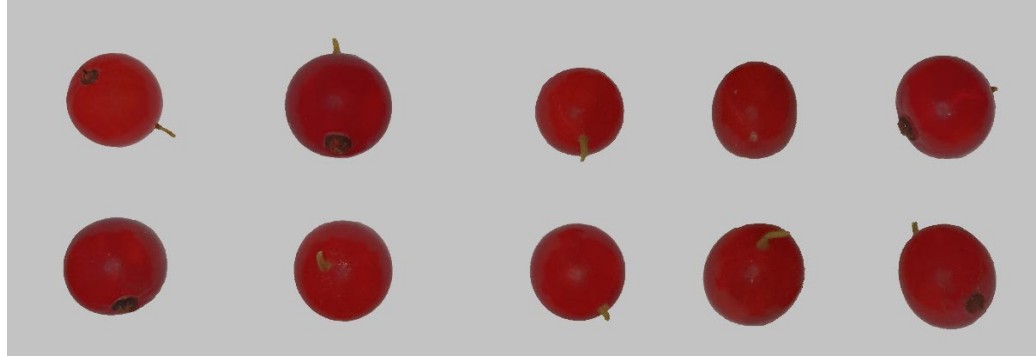

**Figure 1.** Exemplary images of unstored red currants at the beginning of the experiments.

The images of samples stored at room temperature for one week and two weeks are shown in Figure 2. The progressive damage to the outer structure of the fruit with more visible dents and wrinkles is noticeable with increasing storage time. Different levels of fruit damage are also visible in the case of red currants stored in a refrigerator for one week and two weeks (Figure 3).

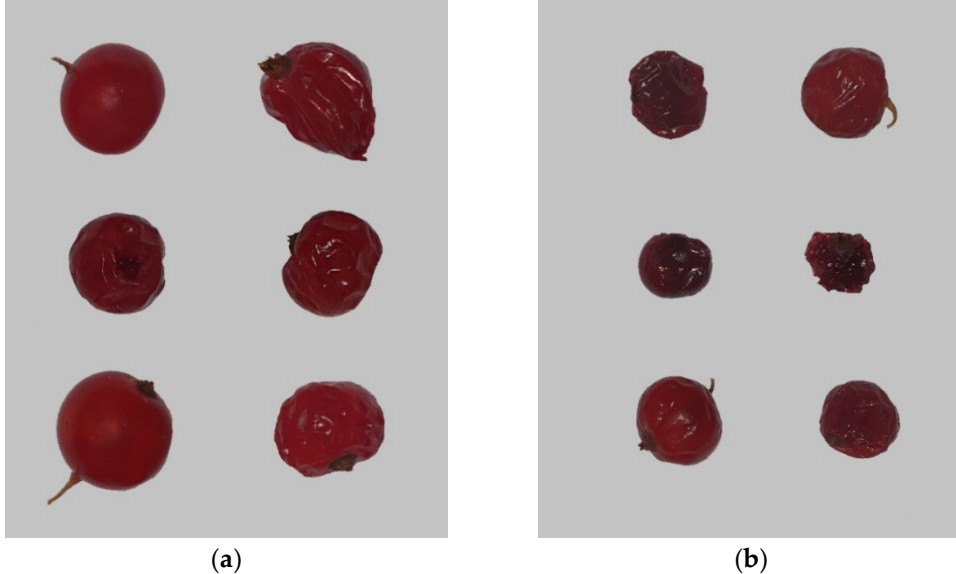

Figure 2. Images of selected red currants stored at room temperature for one week (**a**) and two weeks (**b**).

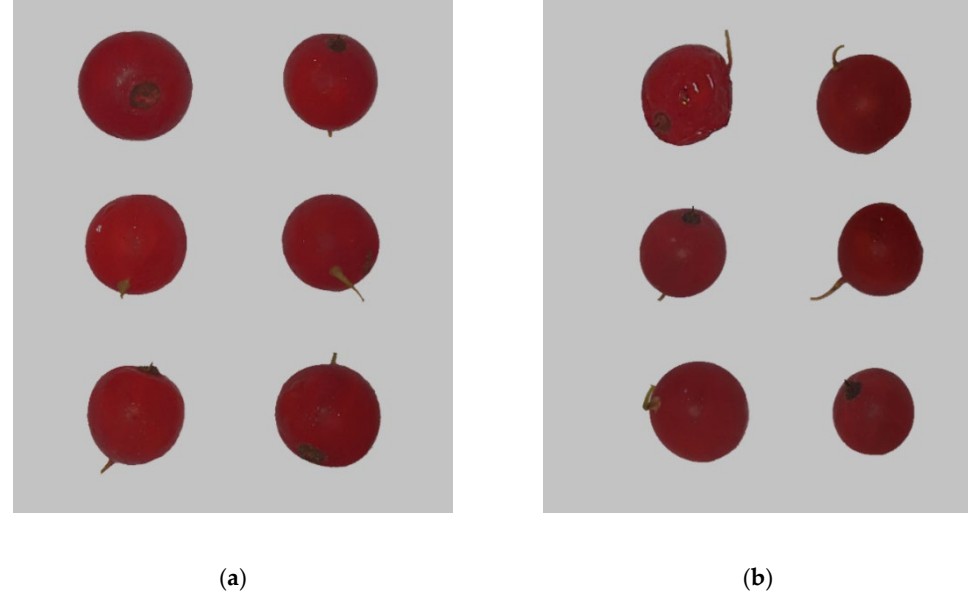

Figure 3. Images of sample red currants stored in the refrigerator for one week (**a**) and two weeks (**b**).

Red currant images were processed using the MaZda software (Łódź University of Technology, Institute of Electronics, Łódź, Poland) [29–31]. Before image processing, the background of the images was changed to black with an intensity of 0. This step was performed to facilitate the segmentation of the images into fruit and background and the determination of the regions of interest (ROIs). The file format of images was changed to BMP. Then, images were converted to different color channels *R*, *G*, *B*, *L*, *a*, *b*, *X*, *Y*, and *Z* using MaZda. Color channels *R* (red), *G* (green), and *B* (blue) belonged to the RGB color space, color channels *L* (lightness from black to white), *a* (red for positive values and green for negative), and *b* (yellow for positive values and blue for negative) were from the Lab color space, and color channels *X* (a component of color information), *Y* (lightness), and *Z* (a component of color information) were from the XYZ color space [32]. The image segmentation into fruit and the background was performed based on the intensity of pixel brightness. The black background had an intensity of 0, whereas each ROI including the

whole red currant was lighter with an intensity greater than 0. The procedure for the color conversion and ROI determination is presented in Figure 4.

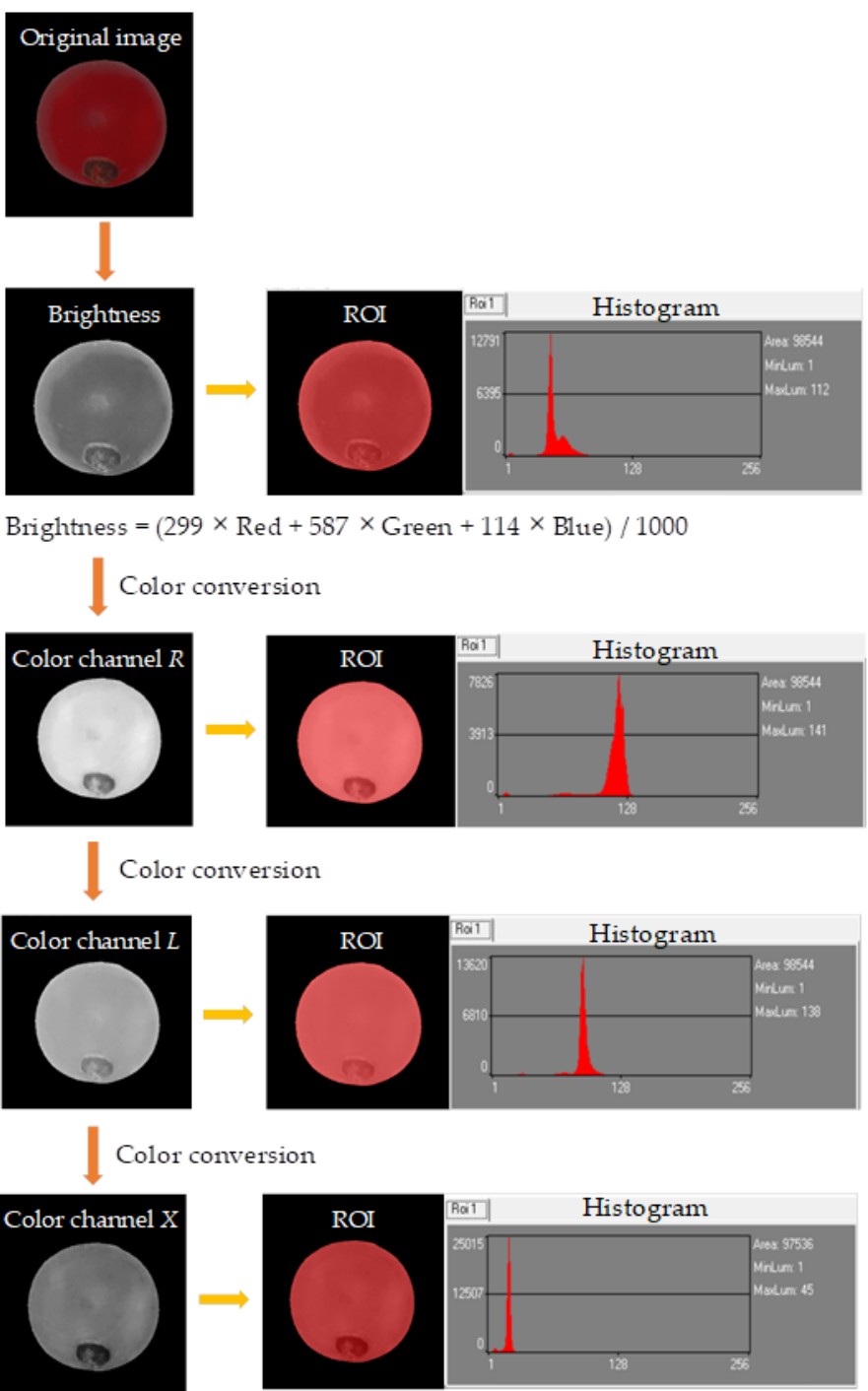

**Figure 4.** The color conversion of the original red currant image to selected color channels *R*, *L*, and *X* and the ROI determination.

The texture information was extracted based on the run-length matrix (parameters: run length nonuniformity, grey level nonuniformity, long run emphasis, short run emphasis, fraction of image in runs for 4 directions), co-occurrence matrix (parameters: angular second moment, contrast, correlation, sum of squares, inverse difference moment, sum average, sum variance, sum entropy, entropy, difference variance, difference entropy for 5 between-pixel distances for 4 directions), gradient map (parameters: absolute gradient

mean, absolute gradient variance, absolute gradient skewness, absolute gradient kurtosis, percentage of pixels with nonzero gradient), histogram (parameters: histogram's mean, histogram's variance, histogram's skewness, histogram's kurtosis, 1% percentile, 10% percentile, 50% percentile, 90% percentile, 99% percentile), Haar wavelet transform (parameters: wavelet energy at 5 scales within four frequency bands), and autoregressive model (parameters: Teta1, Teta2, Teta3, Teta4, Sigma) after transforming ROI images. For each ROI, 1629 texture parameters were determined including 181 textures for each of the color channels *R*, *G*, *B*, *L*, *a*, *b*, *X*, *Y*, and *Z*.

### 2.3. Statistical Analysis

Statistical analysis of obtained data was performed using STATISTICA 13.3 (StatSoft Polska Sp. z o.o., Kraków, Poland) and WEKA software (Machine Learning Group, University of Waikato). STATISTICA software was used to perform the mean comparison of selected textures from each color channel of *R*, *G*, *B*, *L*, *a*, *b*, *X*, *Y*, and *Z*. The normality of the distribution was checked using Shapiro–Wilk, Lilliefors, and Kolmogorov–Smirnov tests, and the homogeneity of variance—using Brown-Forsythe and Levene's tests. The means were compared using the Newman-Keuls parametric test at a significance level of $p \leq 0.05$. The differences in parameters of RHMean, GHMean, BHMean, LHMean, aHMean, bHMean, XHMean, YHMean, and ZHMean were analyzed between unstored red currants and red currants stored at room temperature for one week vs. red currants stored at room temperature for two weeks, as well as unstored red currants vs. red currants stored in the refrigerator for one week vs. red currants stored in the refrigerator for two weeks. As a result, the statistically significant influence of storage time on changes in the mean values of selected texture parameters was determined.

The classification of samples was carried out using WEKA software [33,34]. The analysis for four different datasets was performed. Models based on combined selected image textures from color channels *R*, *G*, *B*, *L*, *a*, *b*, *X*, *Y*, and *Z* were built for distinguishing:

(1) unstored red currants intended for storage at room temperature (200 cases), red currants stored at room temperature for one week (200 cases), and red currants stored at room temperature for two weeks (200 cases);

(2) unstored red currants intended for storage in the refrigerator (200 cases), red currants stored in the refrigerator for one week (200 cases), and red currants stored in the refrigerator for two weeks (200 cases);

(3) red currants stored at room temperature for one week (200 cases) and red currants stored in the refrigerator for one week (200 cases);

(4) red currants stored at room temperature for two weeks (200 cases) and red currants stored in the refrigerator for two weeks (200 cases).

In the case of each classification, the textures with the highest discriminative power were selected using the Ranker with OneR attribute evaluator. Selected textures were used to develop models using the machine learning algorithms belonging to different groups, such as JRip (Java repeated incremental pruning) and PART from the group of Rules; J48, Random Forest, and LMT (Logistic Model Tree) from the group of Trees; Logit Boost, Multi Class Classifier, Filtered Classifier, and Random Committee from the group of Meta; Logistic, SMO (Sequential Minimal Optimization), RBF (Radial Basis Function) Classifier and Multilayer Perfection from the group of Functions; and Naive Bayes and Bayes Net from the group of Bayes. The classification of samples was carried out using a test mode of 10-fold cross-validation. For each classification, a dataset was randomly divided into 10 parts, including nine parts as the training sets and one part as the test set. Each of the ten parts was considered as the test set in turn and the remaining nine parts—as the training sets. Thus, the learning was carried out 10 times with different training sets and the average of 10 estimates was calculated. The most effective machine learning algorithms

were selected based on the highest accuracies and values of Precision (Equation (1)), Recall (Equation (2)), and F-Measure (Equation (3)) [25,35].

$$\text{Precision} = TP/(TP + FP) \tag{1}$$

$$\text{Recall} = TP/(TP + FN) \tag{2}$$

$$\text{F−Measure} = 2TP/(2TP + FP + FN) \tag{3}$$

TP: True Positive; FP: False Positive; FN: False Negative.

## 3. Results and Discussion

The influence of storage conditions and time on red currants was assessed in an objective and non-destructive manner considering the statistically significant differences in the mean of the selected texture parameters and the results of the classification of samples performed using models built based on image textures using machine learning algorithms. Firstly, the changes in texture HMean for each color channel, such as RHMean (channel *R*), GHMean (channel *G*), BHMean (channel *B*), LHMean (channel *L*), aHMean (channel *a*), bHMean (channel *b*), XHMean (channel *X*), YHMean (channel *Y*), and ZHMean (channel *Z*), caused by storage time were analyzed separately for samples stored at room temperature (Figure 5) and in the refrigerator (Figure 6). The differences in texture parameters as a result of prolonged storage were observed for both samples stored in the room and the refrigerator. However, despite the same trends, the changes in the red currant samples stored at room temperature were greater, and statistically significant differences between the samples were observed for more parameters.

Red currants stored at room temperature (Figure 5) were characterized by a statistically significant decrease in the values of RHMean, LHMean, aHMean, bHMean, XHMean, YHMean, and ZHMean with increasing storage time. All three red currant samples—unstored, stored at room temperature for one week, and stored at room temperature for two weeks—formed separate homogeneous groups. Meanwhile, in the case of GHMean and BHMean, an increase in the values was observed with increasing time of red currant storage. For the texture of GHMean, each sample was included in a separate homogenous group, and in the case of BHMean, samples stored at room temperature for one week and two weeks formed one homogenous group with statistically significantly higher values than for unstored red currants.

In the case of red currants stored in the refrigerator (Figure 6), a decrease in the values of RHMean, LHMean, aHMean, bHMean, XHMean, YHMean, and ZHMean and an increase in the values of GHMean and BHMean with increasing storage time were also observed. However, unstored red currants and samples stored in the refrigerator for one week and two weeks created three separate homogenous groups with statistically significantly different values only in the case of aHMean, bHMean, XHMean, and GHMean. For BHMean, storage of red currant in the refrigerator for one week resulted in a statistically significant increase in the value of this texture, and the extension of the storage time did not cause any further significant changes. In the case of red currants stored in the refrigerator for one week, no statistically significant changes in RHMean, LHMean, and YHMean were observed compared to the unstored sample examined at the beginning of the experiment. For the ZHMean texture, no statistically significant differences were found, and all three samples were included in one homogenous group. The obtained results may indicate that changes in the external structure of red currants manifested in changes in the values of the textures of the outer surface were smaller for the samples stored in the refrigerator (Figure 6) than for the samples stored at room temperature (Figure 5).

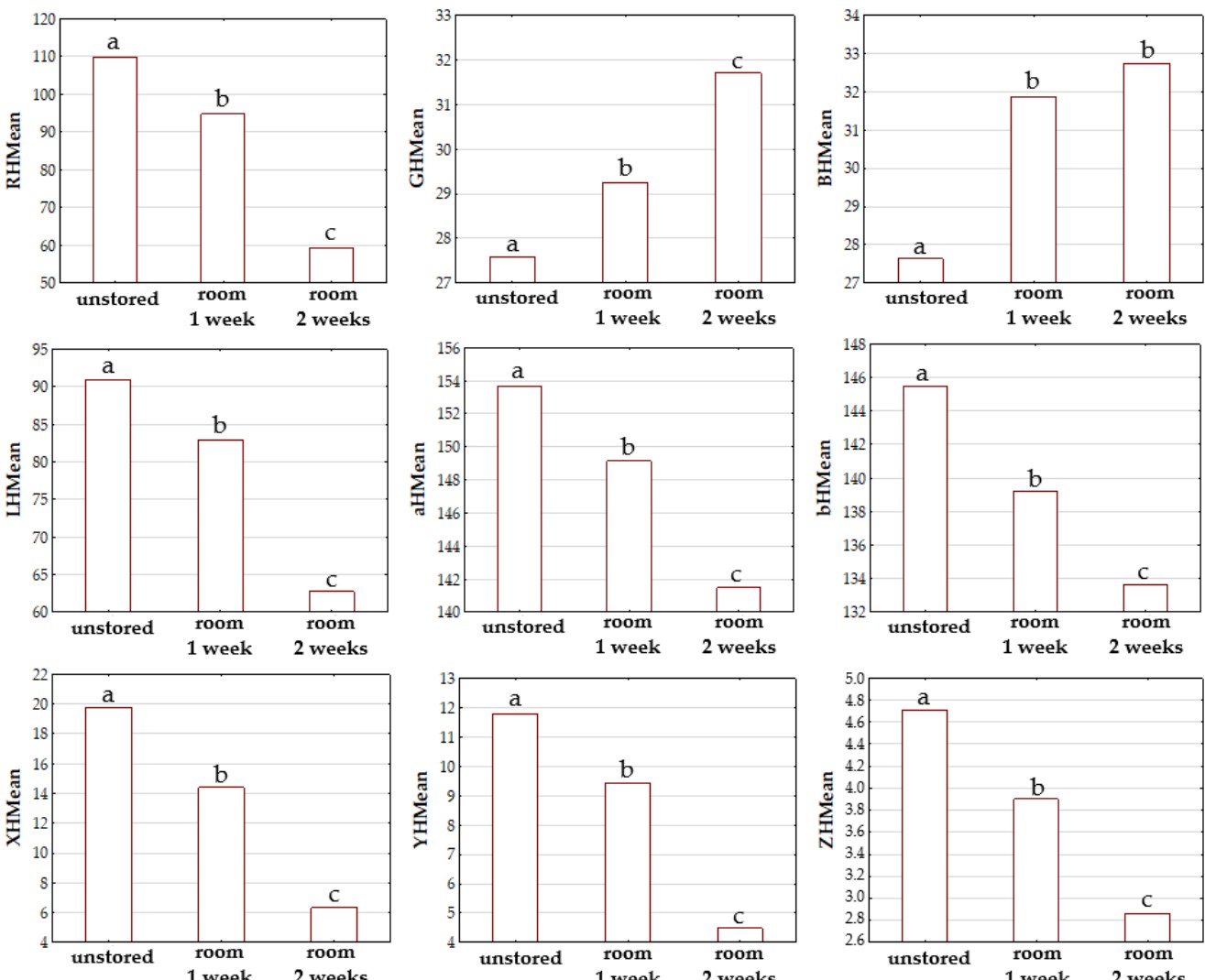

**Figure 5.** The changes in the values of selected image textures of red currants stored at room temperature. a, b, c—the same letters on one graph denote no statistical differences between samples.

Distinguishing samples using models based on image textures developed using various traditional machine learning algorithms also revealed the influence of both storage conditions and time on the external structure of red currants. This paper presents the results for selected algorithms that provided the most satisfactory results. In the case of the storage of red currants at room temperature (Table 1), an average accuracy of 100%, and the values of Precision, Recall and F-Measure equal to 1.000 for each class (unstored, stored for one week and stored for two weeks) were achieved for most of the applied algorithms. In the case of each group of machine learning algorithms, examined samples were completely correctly classified using selected algorithms, such as PART from the group of Rules, Random Forest from the group of Trees, Multi Class Classifier from the group of Meta, SMO from the group of Functions, and Naive Bayes from the group of Bayes. By using these algorithms to build models, all 200 cases of unstored red currants were correctly classified as unstored ones, all 200 cases of red currants stored at room temperature for one week were correctly classified as stored at room temperature for one week, and all 200 cases from the actual class of samples stored at room temperature for two weeks were correctly included in the predicted class of red currants stored at room temperature for two weeks. In the case of the RBF Classifier from the group of Functions, an average accuracy of 99% was observed. The unstored red currants were completely correctly (100%) distinguished from other classes and the values of Precision, Recall and F-Measure of 1.000 were determined.

For both classes of red currants stored at room temperature for one week and two weeks, an accuracy of 98% and Precision, Recall and F-Measure of 0.980 were found. Overall, 2% of cases belonging to the actual class of red currants stored at room temperature for one week were incorrectly included in the class of red currants stored at room temperature for two weeks, and 2% of cases from the actual class of samples stored at room temperature for two weeks were incorrectly classified as red currants stored at room temperature for one week. Very high correctness of classification reaching 100% and the values of Precision, Recall and F-Measure of up to 1.000 for most algorithms proved the great influence of the storage time on the textures of the outer surface of the red currants stored at room temperature.

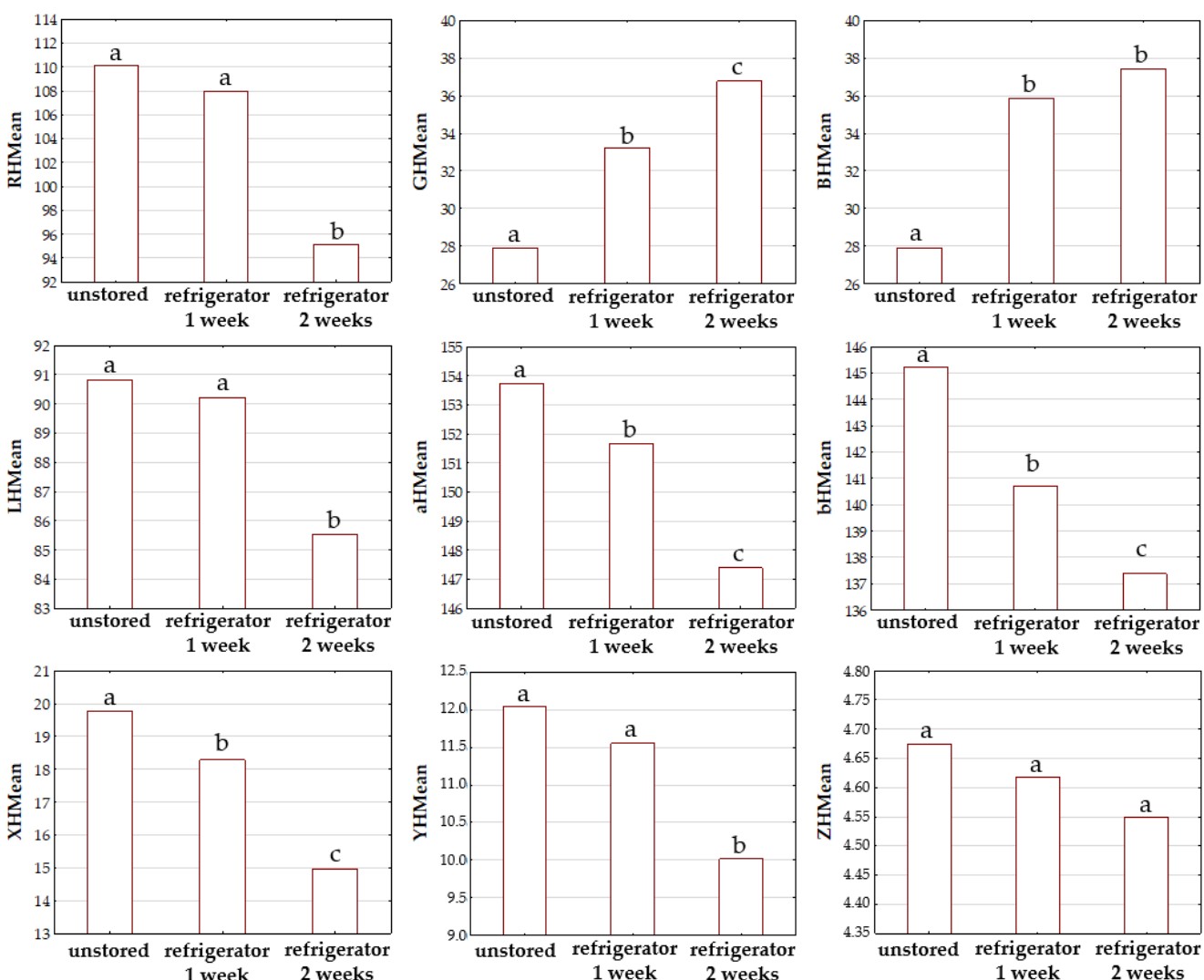

**Figure 6.** The differences in selected image texture parameters of red currants stored in the refrigerator caused by storage time. a, b, c—the same letters on one graph denote no statistical differences between samples.

**Table 1.** The results of the classification of unstored red currants, and fruit stored at room temperature for one week, and stored at room temperature for two weeks based on models combining selected textures from color channels *R*, *G*, *B*, *L*, *a*, *b*, *X*, *Y*, and *Z* developed using machine learning algorithms from different groups.

| Algorithm | Predicted Class (%) | | | Actual Class | Average Accuracy (%) | Precision | Recall | F-Measure |
|---|---|---|---|---|---|---|---|---|
| | Unstored | Room 1 Week | Room 2 Weeks | | | | | |
| PART (Rules) | 100 | 0 | 0 | unstored | 100 | 1.000 | 1.000 | 1.000 |
| | 0 | 100 | 0 | room 1 week | | 1.000 | 1.000 | 1.000 |
| | 0 | 0 | 100 | room 2 weeks | | 1.000 | 1.000 | 1.000 |
| Random Forest (Trees) | 100 | 0 | 0 | unstored | 100 | 1.000 | 1.000 | 1.000 |
| | 0 | 100 | 0 | room 1 week | | 1.000 | 1.000 | 1.000 |
| | 0 | 0 | 100 | room 2 weeks | | 1.000 | 1.000 | 1.000 |
| Multi Class Classifier (Meta) | 100 | 0 | 0 | unstored | 100 | 1.000 | 1.000 | 1.000 |
| | 0 | 100 | 0 | room 1 week | | 1.000 | 1.000 | 1.000 |
| | 0 | 0 | 100 | room 2 weeks | | 1.000 | 1.000 | 1.000 |
| RBF Classifier (Functions) | 100 | 0 | 0 | unstored | 99 | 1.000 | 1.000 | 1.000 |
| | 0 | 98 | 2 | room 1 week | | 0.980 | 0.980 | 0.980 |
| | 0 | 2 | 98 | room 2 weeks | | 0.980 | 0.980 | 0.980 |
| SMO (Functions) | 100 | 0 | 0 | unstored | 100 | 1.000 | 1.000 | 1.000 |
| | 0 | 100 | 0 | room 1 week | | 1.000 | 1.000 | 1.000 |
| | 0 | 0 | 100 | room 2 weeks | | 1.000 | 1.000 | 1.000 |
| Naive Bayes (Bayes) | 100 | 0 | 0 | unstored | 100 | 1.000 | 1.000 | 1.000 |
| | 0 | 100 | 0 | room 1 week | | 1.000 | 1.000 | 1.000 |
| | 0 | 0 | 100 | room 2 weeks | | 1.000 | 1.000 | 1.000 |

Storage in the refrigerator resulted in lower correctness of the classification of unstored red currants, and samples stored for one week, and stored for two weeks for some algorithms (Table 2) than in the case of samples stored at room temperature (Table 1). It may indicate less noticeable changes in the structure of the fruit caused by storage at a low temperature in the refrigerator. In the case of the storage in the refrigerator (Table 2), the average accuracy of 100% and the Precision, Recall and F-Measure of 1.000 were only obtained for the models built using two (SMO and Naive Bayes) out of seven machine learning algorithms. An average accuracy equal to 99% was observed for models built using Random Forest and Multi Class Classifier. However, in the case of the Random Forest algorithm, unstored red currants and those stored in the refrigerator for one week were distinguished from each other and fruit stored in the refrigerator for two weeks with accuracies of 100%, whereas for Multi Class Classifier, only unstored red currants were correctly classified in 100% of cases. In the case of other machine learning algorithms, an accuracy of 98% was found for RBF Classifier and 97% for PART.

**Table 2.** The performance metrics of the classification of unstored red currants, and samples stored in the refrigerator for one week, and stored in the refrigerator for two weeks using models built using various machine learning algorithms based on combined selected textures from color channels *R*, *G*, *B*, *L*, *a*, *b*, *X*, *Y*, and *Z*.

| Algorithm | Predicted Class (%) | | | Actual Class | Average Accuracy (%) | Precision | Recall | F-Measure |
|---|---|---|---|---|---|---|---|---|
| | Unstored | Refrigerator 1 Week | Refrigerator 2 Weeks | | | | | |
| PART (Rules) | 96 | 4 | 0 | unstored | 97 | 1.000 | 0.960 | 0.980 |
| | 0 | 100 | 0 | refrigerator 1 week | | 0.926 | 1.000 | 0.962 |
| | 0 | 4 | 96 | refrigerator 2 weeks | | 1.000 | 0.960 | 0.980 |
| Random Forest (Trees) | 100 | 0 | 0 | unstored | 99 | 1.000 | 1.000 | 1.000 |
| | 0 | 100 | 0 | refrigerator 1 week | | 0.980 | 1.000 | 0.990 |
| | 0 | 2 | 98 | refrigerator 2 weeks | | 1.000 | 0.980 | 0.990 |
| Multi Class Classifier (Meta) | 100 | 0 | 0 | unstored | 99 | 0.980 | 1.000 | 0.990 |
| | 2 | 98 | 0 | refrigerator 1 week | | 0.980 | 0.980 | 0.980 |
| | 0 | 2 | 98 | refrigerator 2 weeks | | 1.000 | 0.980 | 0.990 |
| RBF Classifier (Functions) | 100 | 0 | 0 | unstored | 98 | 0.980 | 1.000 | 0.990 |
| | 2 | 96 | 2 | refrigerator 1 week | | 0.980 | 0.960 | 0.970 |
| | 0 | 2 | 98 | refrigerator 2 weeks | | 0.980 | 0.980 | 0.980 |
| SMO (Functions) | 100 | 0 | 0 | unstored | 100 | 1.000 | 1.000 | 1.000 |
| | 0 | 100 | 0 | refrigerator 1 week | | 1.000 | 1.000 | 1.000 |
| | 0 | 0 | 100 | refrigerator 2 weeks | | 1.000 | 1.000 | 1.000 |
| Naive Bayes (Bayes) | 100 | 0 | 0 | start | 100 | 1.000 | 1.000 | 1.000 |
| | 0 | 100 | 0 | refrigerator 1 week | | 1.000 | 1.000 | 1.000 |
| | 0 | 0 | 100 | refrigerator 2 weeks | | 1.000 | 1.000 | 1.000 |

After revealing that time affects the textures of the images of red currants (Tables 1 and 2), in the next stages of analysis, the influence of storage conditions was carefully assessed (Tables 3 and 4). The differences in selected textures from color channels *R*, *G*, *B*, *L*, *a*, *b*, *X*, *Y*, and *Z* allowed for building models distinguishing red currants stored at room temperature for one week and stored in the refrigerator for one week with an accuracy of 100% and the Precision, Recall, and F-Measure of 1.000 for the PART, Random Forest, Multi Class Classifier, RBF Classifier, and SMO machine learning algorithms and 99% for Naive Bayes (Table 3). In the case of Naive Bayes, the accuracy for red currants stored at room temperature for one week was 100% and for samples stored in the refrigerator for one week—98% and the remaining 2% of cases were classified as stored at room temperature. The values of Precision (0.980 for samples stored in the room and 1.000 for samples stored in the refrigerator), Recall (1.000 for samples stored in the room and 0.980 for samples stored in the refrigerator), and F-Measure (0.990 for both samples) were also high.

**Table 3.** The accuracies and other performance metrics for classifying red currants stored at room temperature for one week and stored in the refrigerator for one week based on selected textures from color channels *R*, *G*, *B*, *L*, *a*, *b*, *X*, *Y*, and *Z*.

| Algorithm | Predicted Class (%) | | Actual Class | Average Accuracy (%) | Precision | Recall | F-Measure |
|---|---|---|---|---|---|---|---|
| | Room 1 Week | Refrigerator 1 Week | | | | | |
| PART Random Forest Multi Class Classifier RBF Classifier | 100 | 0 | room 1 week | 100 | 1.000 | 1.000 | 1.000 |
| | 0 | 100 | refrigerator 1 week | | 1.000 | 1.000 | 1.000 |
| SMO Naive Bayes | 100 | 0 | room 1 week | 99 | 0.980 | 1.000 | 0.990 |
| | 2 | 98 | refrigerator 1 week | | 1.000 | 0.980 | 0.990 |

**Table 4.** The confusion matrices, average accuracies and the values of Precision, Recall and F-Measure of the classification of red currants stored at room temperature for two weeks and stored in the refrigerator for two weeks for models including selected textures from color channels *R*, *G*, *B*, *L*, *a*, *b*, *X*, *Y*, and *Z*.

| Algorithm | Predicted Class (%) | | Actual Class | Average Accuracy (%) | Precision | Recall | F-Measure |
|---|---|---|---|---|---|---|---|
| | Room 2 Weeks | Refrigerator 2 Weeks | | | | | |
| PART Random Forest Multi Class Classifier RBF Classifier SMO Naive Bayes | 100 | 0 | room 2 weeks | 100 | 1.000 | 1.000 | 1.000 |
| | 0 | 100 | refrigerator 2 weeks | | 1.000 | 1.000 | 1.000 |

Red currants stored at room temperature for two weeks and stored in the refrigerator for two weeks were completely different in terms of the selected textures of the outer surface (Table 4). All applied machine learning algorithms of PART, Random Forest, Multi Class Classifier, RBF Classifier, SMO, and Naive Bayes distinguished both samples with an accuracy of 100% and Precision, Recall, and F-Measure of 1.000.

The combination of image processing and machine learning proved to be effective to monitor the changes in red currants during storage. The obtained results may be of great practical importance. Due to the short shelf life, currants are available in fresh form for a short time in the year. Storage, especially at a lower temperature or freezing, can allow for extending their shelf life [36,37]. The significance of the present work is related to developing innovative models using image features to detect the changes in red currant quality during storage. Due to the use of image analysis and artificial intelligence, even slight changes were identified in a non-destructive, easy, fast, and inexpensive manner with high accuracy. The correct detection of the changes in fruit structure during storage can contribute to the selection of stored red currants with the desired properties for consumption or processing and the rejection of unusable fruit.

Nowadays, the demand for high-quality fruit products and automatic high-throughput quality detection is increasing. Extending the shelf life of berries can provide new options to producers and consumers. The detection and prediction of berry quality using advanced artificial intelligence-based techniques can be considered the important direction of modern food processing. However, techniques involving image processing and artificial intelligence are not commonly used in all aspects of commercial berry preservation [38]. Therefore, the undertaken own study expanded the scope of the application of the approach combining image processing and artificial intelligence and indicated new research directions for developed procedures of fruit quality monitoring. Additionally, in the case of a decrease in the number of farmers, the workforce can be replaced by technology [39]. Furthermore,

machine vision can ensure quicker and more accurate identification of quality changes in berries than manual evaluation [40]. Manual inspection is more laborious, error-prone, and time-consuming than non-destructive imaging techniques based on pattern recognition to assess the berry damages [41]. The importance of classifying fruit samples based on their external quality parameters using imaging and machine learning models is great for agroindustry and agribusiness [42]. In the present study, the classification of red currant berries was assessed considering the accuracy, Precision, Recall, and F-Measure for the models developed based on image textures using machine learning algorithms. This increases the importance of the research carried out. In practical applications, the identification of damaged berries using Recall, Precision, and F-Measure is rarely used by investigators engaged in agricultural engineering. However, these evaluation indicators are very important. Considering only the classification accuracy is not enough. Using more performance metrics can contribute to making more effective decisions and reduce economic losses [43]. The obtained own results confirmed that the influence of storage technologies on the external structure of red currants can be assessed using image analysis and machine learning. This could allow the development of robust models to predict the maintenance of optimal fruit structure during storage under various conditions. The developed procedures can be useful for farmers and food processors who are unable to process all raw materials at once and need to store them before processing. The proposed approach can be used in practice to develop vision systems to predict changes in the quality of stored red currants and to assess the suitability of stored raw materials for consumer consumption and processing.

### 4. Conclusions

Image analysis combined with artificial intelligence proved to be effective to monitor the changes in stored red currants. The use of models based on features from different color channels *R*, *G*, *B*, *L*, *a*, *b*, *X*, *Y*, and *Z* of digital images developed using various machine learning algorithms for the detection of fruit quality during storage is a great novelty of this study and allowed for performing research in an objective, non-destructive and effective manner. Statistically significant changes in selected texture parameters of the outer surface of red currants as a result of increasing storage time were found for samples stored at room temperature and in the refrigerator. Using machine learning models, the completely correct classification (100% accuracies) of samples stored under different conditions was achieved. Due to the promising results, further research could be undertaken to develop vision systems for predicting quality changes in stored red currants and assessing the suitability of stored raw materials for consumer consumption and processing.

**Funding:** This research received no external funding.

**Institutional Review Board Statement:** Not applicable.

**Data Availability Statement:** The data presented in this study are available on request from the corresponding author.

**Conflicts of Interest:** The authors declare no conflict of interest.

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
