# Peer review of "Assessment of the Influence of Storage Conditions and Time on Red Currants (Ribes rubrum L.) Using Image Processing and Traditional Machine Learning"

_agriculture, doi:10.3390/agriculture12101730_

Round 1

Reviewer 1 Report

The authors present a statistical analysis of images of red currants and how the different methods of storage affect the visual quality of the product. In my capacity as an image processing expert, I should point out that there is no novelty from the image processing point of view. Nevertheless, I acknowledge that the study might be of interest for the readership of the journal. 

Keeping the above in mind, the author should enhance the paper so as to clearly present the methodology being followed. Right now in the text we can only find a very brief description of how the ROIs have been treated. Yet, for the average reader of this journal, this description might not be sufficient, in order to reproduce the results. A scientific paper should contain all the adequate details of reproducibility and thus, the author is advised to extend paragraph 2.2 to sufficiently include such details.

A minor comment: R,G,B,L,a,b,X,Y,Z contain colour information and not texture. Only after the wavelet transform is applied on a ROI, texture information is available. 

Author Response

Reviewer 1

The authors present a statistical analysis of images of red currants and how the different methods of storage affect the visual quality of the product. In my capacity as an image processing expert, I should point out that there is no novelty from the image processing point of view. Nevertheless, I acknowledge that the study might be of interest for the readership of the journal. 

Response: Thank you very much for your careful reading of the manuscript and this comment. I agree with this opinion. Information about image processing and machine learning techniques is available in the previous literature. However, there is no literature data on the use of this approach to classify red currants subjected to various storage conditions. The innovative approach is related to the field of agriculture. Therefore, the manuscript has been sent to this journal. The novelty of this study is the expansion of the scope of the techniques used and indicating new research directions for image processing and artificial intelligence.

Keeping the above in mind, the author should enhance the paper so as to clearly present the methodology being followed. Right now in the text we can only find a very brief description of how the ROIs have been treated. Yet, for the average reader of this journal, this description might not be sufficient, in order to reproduce the results. A scientific paper should contain all the adequate details of reproducibility and thus, the author is advised to extend paragraph 2.2 to sufficiently include such details.

Response: Thank you very much for this valuable comment. Subsection 2.2. Image analysis has been significantly improved and expanded. More details have been added as follows:

“Red currant images were processed using the MaZda software (Łódź University of Technology, Institute of Electronics, Łódź, Poland) [29-31]. Before image processing, the background of the images was changed to black with an intensity of 0. This step was performed to facilitate the segmentation of the images into fruit and background and the determination of the regions of interest (ROIs). The file format of images was changed to BMP. Then, images were converted to different color channels R, G, B, L, a, b, X, Y, and Z using MaZda. Color channels R (red), G (green), and B (blue) belonged to the RGB color space, color channels L (lightness from black to white), a (red for positive values and green for negative), and b (yellow for positive values and blue for negative) were from the Lab color space, and color channels X (a component of color information), Y (lightness), and Z (a component of color information) were from the XYZ color space [32]. The image segmentation into fruit and the background was performed based on the intensity of pixel brightness. The black background had an intensity of 0. Whereas each ROI including the whole red currant was lighter with an intensity greater than 0. The procedure for the color conversion and ROI determination is presented in Figure 4.”

Figure 4. The color conversion of the original red currant image to selected color channels R, L, and X and the ROI determination.

  1. Szczypiński, P.M.; Strzelecki, M.; Materka, A.; Klepaczko, A. MaZda—a software package for image texture analysis. Comput Methods Progr Biomed 2009, 94(1), 66–76.
  2. Szczypiński, P.M.; Strzelecki, M.; Materka, A. MaZda—a software for texture analysis. In: Proceedings of ISITC 2007, November 23–23, 2007, Republic of Korea 2007, 245–249.
  3. Strzelecki, M.; Szczypiński, P.; Materka, A.; Klepaczko, A. A software tool for automatic classification and segmentation of 2D/3Dmedical images. Nucl Inst Methods Phys Res A 2013, 702, 137–140.
  4. Ibraheem, N.A.; Hasan, M.M.; Khan, R.Z.; Mishra, P.K. Understanding color models: A review. ARPN J. Sci. Technol. 2012, I, 265–275.

A minor comment: R,G,B,L,a,b,X,Y,Z contain colour information and not texture. Only after the wavelet transform is applied on a ROI, texture information is available. 

Response: I am grateful to the Reviewer for this detailed comment. It seemed obvious. However, it has been specified as follows: “The texture information was extracted based on the run-length matrix, co-occurrence matrix, gradient map, histogram, Haar wavelet transform, and autoregressive model after transforming ROI images. For each ROI, 1629 texture parameters were determined including 181 textures for each of the color channels R, G, B, L, a, b, X, Y, and Z.”

Reviewer 2 Report

The manuscript submitted for review (Agriculture-1966271) is of particular scientific interest. However, the manuscript needs improvement:

1.                  Section 2.1. is not fully presented

- It is necessary to specify the cultivar that was tested in this work, the genetic origin of this cultivar

- How (visually or technically) was the assessment of the removable maturity of red currant berries carried out? What criteria were used to assess the full maturity of berries, how were the berries selected for storage?  See the work of DOI 10.17221/11/2020-RAE

- Based on what did the author decide to store berries, not bunches?

- Were the boxes used in the experiment hermetically packed?

- Indicate which primary signs of berry changes served as indicators for stopping the experiment

- Specify the brand of refrigeration equipment that the authors used in the experiment

2 It is necessary to add information in the "Results" section and emphasize the theoretical significance of your work (biological)

3. The "Discussion of Results" information is poorly presented. And the provided information does not really relate to the results of the author's work

Indicate in the final part of the paper how the research results can be applied in practice for farmers.

Author Response

Reviewer 2

The manuscript submitted for review (Agriculture-1966271) is of particular scientific interest.

Response: I am grateful to the Reviewer for this comment.

However, the manuscript needs improvement:

  1. Section 2.1. is not fully presented

Response: Section 2.1 has been completely changed. It has been significantly improved. A lot of new information has been added.

- It is necessary to specify the cultivar that was tested in this work, the genetic origin of this cultivar

Response: It has been described in detail as follows: “The red currants were collected from the local garden located in northeastern Poland. Red currants have been grown in the garden for several years. This study generally aimed to demonstrate the usefulness of image analysis and artificial intelligence in assessing the quality of stored red currant. Therefore, one random cultivar was used in the experiments without considering the characteristics and genetic origin of this cultivar. The only visual criterion for selecting a cultivar was large, fully developed fruit in a given growing season.”

- How (visually or technically) was the assessment of the removable maturity of red currant berries carried out? What criteria were used to assess the full maturity of berries, how were the berries selected for storage?  See the work of DOI 10.17221/11/2020-RAE

Response: The following sentence has been added: “The red currants were harvested at the stage in which fruit were richly colored, juicy, and firm. These features were assessed organoleptically.”

The work of DOI 10.17221/11/2020-RAE also reported the visual determination of ripening and characteristic color of fruit.

- Based on what did the author decide to store berries, not bunches?

Response: It has been specified as follows: “From each bunch of red currants, several undamaged fruits with stems were sampled, to obtain a total of 400 fruits. In the storage experiments, fruit with stems was used so that the structure of the fruit was not damaged during the removal of fruit from the bunches. Individual fruits separated from the bunches were used to ensure the same storage conditions for each fruit. Therefore, red currants were stored as a single layer of fruit. Fruit storage was carried out in plastic boxes with perforated walls.”.

In addition, during harvesting, especially for combine harvesting, some of the currants are also separated from the stems.

- Were the boxes used in the experiment hermetically packed?

Response: It has been specified as follows: “Fruit storage was carried out in plastic boxes with perforated walls.”. The boxes used in the experiment were not hermetically packed.

- Indicate which primary signs of berry changes served as indicators for stopping the experiment

Response: It has been described in more detail as follows: “After two weeks, when very distinct changes in the appearance (size and shape changes such as wrinkling, visible losses of mass and water, and color changes such as fruit darkening) of fruit stored in the room were noticed, indicating complete damage to the structure of most of the red currants, the experiment was stopped.”

- Specify the brand of refrigeration equipment that the authors used in the experiment

Response: It has been specified as follows: “refrigerator (Beko, Istanbul, Turkey)”.

2 It is necessary to add information in the "Results" section and emphasize the theoretical significance of your work (biological)

Response: Thank you very much for this comment.

It has been specified as follows: “The significance of the present work is related to developing innovative models using image features to detect the changes in red currant quality during storage. Due to the use of image analysis and artificial intelligence, even slight changes were identified in a non-destructive, easy, fast, and inexpensive manner with high accuracy. The correct detection of the changes in fruit structure during storage can contribute to the selection of stored red currants with the desired properties for consumption or processing and the rejection of unusable fruit.”

  1. The "Discussion of Results" information is poorly presented. And the provided information does not really relate to the results of the author's work

Response: A completely new Discussion has been written as follows: “The combination of image processing and machine learning proved to be effective to monitor the changes in red currants during storage. The obtained results may be of great practical importance. Due to the short shelf life, currants are available in fresh form for a short time in the year. Storage, especially at a lower temperature or freezing, can allow for extending their shelf life [36,37]. The significance of the present work is related to developing innovative models using image features to detect the changes in red currant quality during storage. Due to the use of image analysis and artificial intelligence, even slight changes were identified in a non-destructive, easy, fast, and inexpensive manner with high accuracy. The correct detection of the changes in fruit structure during storage can contribute to the selection of stored red currants with the desired properties for consumption or processing and the rejection of unusable fruit.

Nowadays, the demand for high-quality fruit products and automatic high-throughput quality detection is increasing. Extending the shelf life of berries can provide new options to producers and consumers. The detection and prediction of berry quality using advanced artificial intelligence-based techniques can be considered the important direction of modern food processing. However, techniques involving image processing and artificial intelligence are not commonly used in all aspects of commercial berry preservation [38]. Therefore, the undertaken own study expanded the scope of the application of the approach combining image processing and artificial intelligence and indicated new research directions for developed procedures of fruit quality monitoring. Additionally, in the case of a decrease in the number of farmers, the workforce can be replaced by technology [39]. Furthermore, machine vision can ensure quicker and more accurate identification of quality changes in berries than manual evaluation [40]. Manual inspection is more laborious, error-prone, and time-consuming than non-destructive imaging techniques based on pattern recognition to assess the berry damages [41]. The importance of classifying fruit samples based on their external quality parameters using imaging and machine learning models is great for agro-industry and agribusiness [42]. In the present study, the classification of red currant berries was assessed considering the accuracy, Precision, Recall, and F-Measure for the models developed based on image textures using machine learning algorithms. This increases the importance of the research carried out. In practical applications, the identification of damaged berries using Recall, Precision, and F-Measure is rarely used by investigators engaged in agricultural engineering. However, these evaluation indicators are very important. Considering only the classification accuracy is not enough. Using more performance metrics can contribute to making more effective decisions and reduce economic losses [43]. The obtained own results confirmed that the influence of storage technologies on the external structure of red currants can be assessed using image analysis and machine learning. This could allow the development of robust models to predict the maintenance of optimal fruit structure during storage under various conditions. The developed procedures can be useful for farmers and food processors who are unable to process all raw materials at once and need to store them before processing. The proposed approach can be used in practice to develop vision systems to predict changes in the quality of stored red currants and to assess the suitability of stored raw materials for consumer consumption and processing.”

Indicate in the final part of the paper how the research results can be applied in practice for farmers.

Response:  It has been indicated as follows: “The obtained own results confirmed that the influence of storage technologies on the external structure of red currants can be assessed using image analysis and machine learning. This could allow the development of robust models to predict the maintenance of optimal fruit structure during storage under various conditions. The developed procedures can be useful for farmers and food processors who are unable to process all raw materials at once and need to store them before processing. The proposed approach can be used in practice to develop vision systems to predict changes in the quality of stored red currants and to assess the suitability of stored raw materials for consumer consumption and processing.”

Round 2

Reviewer 1 Report

The author are invited to report the parameters for the feature extraction methods they have used (e.g. gradient map, histogram, Haar wavelet transform, etc).

Author Response

Thank you very much for this valuable comment. It has been added as follows:

"The texture information was extracted based on the run-length matrix (parameters: run length nonuniformity, grey level nonuniformity, long run emphasis, short run emphasis, fraction of image in runs for 4 directions), co-occurrence matrix (parameters: angular second moment, contrast, correlation, sum of squares, inverse difference moment, sum average, sum variance, sum entropy, entropy, difference variance, difference entropy for 5 between-pixels distances for 4 directions), gradient map (parameters: absolute gradient mean, absolute gradient variance, absolute gradient skewness, absolute gradient kurtosis, percentage of pixels with nonzero gradient), histogram (parameters: histogram’s mean, histogram’s variance, histogram’s skewness, histogram’s kurtosis, 1% percentile, 10% percentile, 50% percentile, 90% percentile, 99% percentile), Haar wavelet transform (parameters: wavelet energy at 5 scales within four frequency bands), and autoregressive model (parameters: Teta1, Teta2, Teta3, Teta4, Sigma) after transforming ROI images. For each ROI, 1629 texture parameters were determined including 181 textures for each of the color channels R, G, B, L, a, b, X, Y, and Z."

Reviewer 2 Report

I do not have any comment. Manuscript can be accepted for publication.

Author Response

Thank you very much for this comment